# The Leading Role of Brain and Abdominal Radiological Features in the Work-Up of Anti-NMDAR Encephalitis in Children: An Up-To-Date Review

**DOI:** 10.3390/brainsci13040662

**Published:** 2023-04-15

**Authors:** Miriana Guarino, Saverio La Bella, Marco Santoro, Daniele Caposiena, Enza Di Lembo, Francesco Chiarelli, Giovanni Iannetti

**Affiliations:** 1Department of Pediatrics, University of Chieti-Pescara “G. D’Annunzio”, Via Dei Vestini 5, Ospedale Clinicizzato Chieti (CH), 66100 Chieti, Italy; saveriolabella@outlook.it (S.L.B.); chiarelli@unich.it (F.C.); 2Department of Radiology, Pescara Public Hospital “Santo Spirito”, 65124 Pescara, Italy; dott.santoromarco@gmail.com (M.S.); dottorcaposiena@gmail.com (D.C.); 3Department of Internist Ultrasound, Pescara Public Hospital “Santo Spirito”, 65124 Pescara, Italy; enza.dilembo@asl.pe.it (E.D.L.); giovanni.iannetti@asl.pe.it (G.I.)

**Keywords:** NMDARe, encephalitis, autoimmune encephalitis, ultrasound, abdominal US

## Abstract

Anti-N-methyl-D-aspartate receptor (NMDAR) encephalitis (NMDARe) is the most common cause of nonviral encephalitis, mostly affecting young women and adolescents with a strong female predominance (F/M ratio of around 4:1). NMDARe is characterized by the presence of cerebrospinal fluid (CSF) antibodies against NMDARs, even though its pathophysiological mechanisms have not totally been clarified. The clinical phenotype of NMDARe is composed of both severe neurological and neuropsychiatric symptoms, including generalized seizures with desaturations, behavioral abnormalities, and movement disorders. NMDARe is often a paraneoplastic illness, mainly due to the common presence of concomitant ovarian teratomas in young women. Abdominal ultrasonography (US) is a key imaging technique that should always be performed in suspected patients. The timely use of abdominal US and the peculiar radiological features observed in NMDARe may allow for a quick diagnosis and a good prognosis, with rapid improvement after the resection of the tumor and the correct drug therapy.

## 1. Introduction

Anti-N-methyl-D-aspartate receptor (NMDAR) encephalitis (NMDARe) is the most common form of autoimmune encephalitis, mostly affecting young women and adolescents [1,2]. This disease usually has a severe impact on the patient’s life, and its clinical features may include both neuropsychiatric (such as anxiety, behavioural abnormalities, or psychosis) and neurologic manifestations, including headaches, decreased level of consciousness, seizures, and various movement disorders, such as orofacial dyskinesias, chorea, or athetosis [3]. Moreover, sleep reduction and serious autonomic involvement (tachycardia, bradycardia, and hypoventilation) have also been described in NMDARe patients [1,4,5,6]. According to the literature, up to 30–70% of female NMDARe patients have an underlying ovarian teratoma, and its resection represents the first-line treatment to achieve a quick clinical improvement [4]. Nevertheless, patients with no teratoma-associated encephalitis are regarded as having a less favorable response to treatment and a high risk of relapse [7,8]. Autoantibody serology, cerebrospinal fluid (CSF) analysis, and imaging techniques are all helpful in making a diagnosis when a patient’s history is hardly suggestive [9]. Ultrasonography (US) and other imaging methods, such as abdominal computed tomography (CT) and brain magnetic resonance imaging (MRI), are key tools for the correct work-up of NMDARe [4,10,11]. As shown in the relevant brain MRI findings described in detail in this paper, the brain disruption may be relevant in this rare disorder, often allowing serious cognitive and psychiatric morbidity. The purpose of this narrative review is to focus on the peculiar characteristics of imaging in NMDARe, with regard to both brain involvement and that of any tumors associated with the disease. Major advancements have been made in the comprehension and management of this rare form of encephalitis, even though there are still many aspects that need to be completely defined and better understood.

## 2. Definition, Etiology, and Pathophysiology

Autoimmune encephalitis is a group of inflammatory disorders of the central nervous system that most commonly affects children and young adults. The disorder is linked to the presence of autoantibodies against neuronal cell-surface proteins, receptors, and ion channels, but some forms of autoimmune encephalitis currently lack identified antibodies. Autoimmune encephalitis accounts for a significant proportion of all cases of encephalitis, and more than half of the patients who suffer from encephalitis do not have an infectious etiology, as reported in “The California Encephalitis Project” [12]. This disorder of the brain parenchyma often affects the cortical or deep grey matter with or without the involvement of the white matter, the meninges, and the spinal cord. The broad group of autoimmune encephalitis may have a wide clinical spectrum of neuropsychiatric symptoms, such as abnormal movements, psychosis, and seizures, up to a coma [1,7,11]. The presence of different autoantibodies can lead to the same manifestations, such as antibodies against mGluR5, anti-MOG, anti-Hu, anti-Ma, and anti-GAD [12]. To date, NMDARe is the most studied autoimmune encephalitis, caused by neuropsychiatric symptoms and CSF antibodies directed against NMDARs. This subtype of encephalitis has a strong female predominance (F/M ratio of 4:1) and an estimated incidence of about 1.5 per million per year [4]. NMDARe was described for the first time in 2007, and currently more than 1000 cases have been reported, of which about 40% had an onset in childhood or adolescence [4]. NMDAR is a postsynaptic inotropic glutamate receptor, consisting in a ligand-gated cation channel permeable to calcium. The receptor is formed by four subunits: two GluN1 (the so-called obligatory subunit) in combination with two GluN2 or GluN3 subunits. GluN2 binds glutamate or its competitive analogs, while GluN1 and GluN3 bind the obligatory coagonist glycine. NMDAR requires two agonists (glutamate and glycine) and membrane depolarization from an adjacent AMPA receptor to open, while it is blocked by magnesium in a voltage-dependent manner. When the receptor is activated by the agonist, calcium enters the cytoplasm, modifying the intracellular calcium pathway for both physiological and pathophysiological events. The receptors are different due to wide subunit combinations, with different expressions and effects [2]. The pathophysiology is still not completely clear, but several similarities have been noted between the synaptic mechanisms of this disease and those related to schizophrenia [1,13]. Indeed, NMDAR channel blockers, such as ketamine, may cause a whole spectrum of schizophrenia-like symptoms and cognitive abnormalities in previously healthy people, even at low doses [14]. Furthermore, in schizophrenia, there are substantial reductions in the expression of the GluN1 subunit of NMDARs in the prefrontal cortex, and many affected people show positive titres of anti-NMDAR antibodies [14]. Moreover, the pathology of NMDARe is consistent with a paraneoplastic limbic encephalitis. In the hippocampus, there is a consistent reduction in the number of NMDARs with a high density of inflammatory cells [15]. Two confirmed triggers of NMDARe are tumors (mostly ovarian teratomas) and viruses [1]. The hypothetical pathogenesis of ovarian teratoma-related NMDARe consists of the expression of NMDARs on the surface of ovarian teratoma cells. The inflammatory infiltrates in the teratoma tumour comprise CD4+ T cells, CD20+ B cells, plasma cells, autoantibodies against NMDAR, central memory cells, and mature dendritic cells. These ones present antigenic fragments of the NMDAR antigen to CD4+ T cells with their activation and proliferation. This process leads to the differentiation of plasma B cells and the subsequent generation of IgG autoantibodies that cross the blood–brain barrier into the cerebrospinal fluid. The autoantibodies alter the surface of the NMDAR, disrupting its interaction with the synaptic proteins. The main targets of the autoantibodies are the hippocampus and prefrontal cortex of the brain, consisting of the presence of neuropsychiatric symptoms [2,16]. Viral infections, including the herpes simplex virus (HSV), varicella-zoster virus (VZV), measles virus, and others, have been associated with a large number of NMDARe. This trigger can explain the seasonal variability of NMDARe in children [17]. Some studies have suggested an association between NMDARe and the HLA-I allele B*07:02 and HLA-II allele DRB1*16:02 [1]. According to the recent “Updated Diagnostic Criteria for Paraneoplastic Neurologic Syndromes”, NMDARe are considered neurologic syndromes at “intermediate risk”, associated with cancer in 30–70% of cases (mostly ovarian or extraovarian teratomas, such as mediastinal localizations), especially in females between 12 and 45 years of age (50% of all patients) [10,18]. On the other hand, women with ovarian teratomas generally have a low prevalence of anti-NMDAR antibodies, and those who develop autoimmune encephalitis seem to have smaller teratomas than those without NMDARe [10]. Meanwhile, paraneoplastic NMDARe are extremely rare in males and children under the age of 12 (less than 10%) [7,8]. The disease course in children can be severe, and up to 75% of patients may require admission to an intensive care unit, although much progress has been made so far, and up to 85% of children can achieve a good prognosis [4]. In affected patients, the immune response frequently leads to an acute or subacute presentation lasting less than three months, and only a few patients have a chronic form of the disease. Indeed, abrupt onset is unusual for an autoimmune presentation, and in such patients, a vascular aetiology should be taken into consideration instead [19,20]. In paraneoplastic encephalitis, which tends to heal once the cancer is removed, a progressive course is more typical [19,20].

## 3. Clinical Features

The diffuse brain inflammation occurring in NMDARe usually results in a wide spectrum of different clinical phenotypes, often linked to stereotypical symptoms [20]. Young patients may develop a very complex spectrum of signs and symptoms, with a serious impact on their lives, involving both neurological and psychiatric aspects. At the onset, about 90% of patients have prominent psychiatric or behavioural symptoms, making the differential diagnosis from a primary psychiatric disorder difficult [4]. Seizures are common and usually represent the first presentation of the disease in the pediatric population [4]. In a recent case series of both adult and pediatric patients, more than half of them developed one or more of the following: tonic–clonic seizures (79%), focal seizures (74%), focal seizures without impaired awareness (55%), focal seizures with impaired awareness (42%), status epilepticus (35%), and refractory status epilepticus (21%) [5]. Specifically, fever, seizure, mental and behavioral disorders, and decreased consciousness are most frequently observed in NMDARe patients with an underlying teratoma, whereas neuropsychiatric symptoms and headache have been predominantly described in patients without teratoma [21]. Several patients may present with a prodromal headache or a viral-like infection state [4,6]. Clinical features of NMDARe often include neuropsychiatric manifestations such as anxiety, behavioural abnormalities, hallucinations, psychosis, sleep reduction or hypersomnia, memory deficits, decreased level of consciousness, mutism, and autonomic instability (most commonly tachycardia, bradycardia, cardiac pauses, and hypoventilation) [6]. In addition, almost all young patients develop abnormal movements, including orofacial dyskinesias, chorea or athetosis, opisthotonos, and oculogyric crisis. However, a specific phenotype has not been identified, and other rare manifestations may occur, such as self-injuries of the tongue or lips, which are not uncommon in affected young patients [1,3]. Compared to adults, children with NMDARe present more often with seizures, insomnia, irritability, or behavioural symptoms, and, moreover, treatment decisions are usually more aggressive, starting immunotherapy early. Thus, neuropsychological involvement can seriously affect the transition from childhood into adulthood, involving both cognitive and emotional aspects [4,11]. Given its peculiar clinical presentation, NMDARe should be suspected in patients presenting with subacute-onset suggestive neuropsychiatric features.

## 4. Diagnostic Criteria

In children with suspected encephalitis, a prompt work-up is required to first rule out both the infectious forms of encephalitis and systemic/metabolic causes. A multidisciplinary approach is essential, including neurologists, psychiatrists, and infectious disease physicians [20]. A prompt diagnosis is essential to starting the correct therapy because immunotherapies can worsen an infectious disease. A correct differential diagnosis is crucial and should consider a broad spectrum of disorders, such as infections, demyelinating disorders, vascular aetiologies, malignancies, metabolic and mitochondrial disorders, neurologic and rheumatologic diseases, and psychiatric disorders [12]. Specific diagnostic criteria that might help to distinguish between low, moderate, and high clinical suspicion have been proposed. Major presenting features are the presence of seizures, movement disorders, and behavioural changes or psychosis, while minor presenting features are dysautonomia, speech changes, focal neurologic deficits, memory disturbances, and a decreased level of consciousness [22]. Patients with all three major features or one major plus three minor features or with all four minor features trigger high clinical suspicion; the presence of two major features or one major and two minor features indicates a moderate clinical suspicion; and lastly, patients with one major feature or two minor features suggest a low clinical suspicion [12,22]. A brain MRI with contrast should be performed as the first step of the diagnostic process in order to confirm the presence of focal or multifocal brain involvement. Nevertheless, a normal brain MRI could be present in children with NMDARe [23]. Researching pleocytosis and NMDAR antibodies in CSF should be the focus of laboratory tests as the second stage in the diagnostic strategy [19]. Indeed, the diagnosis must be confirmed by the presence of IgG antibodies against the GluN1 subunit of NMDARs in CSF. High levels of IgM and IgA antibodies anti-NMDARs are considered too insensitive and specific for the diagnosis, as well as the presence of anti-NMDAR autoantibodies in blood [24]. To exclude other forms of encephalitis, the CSF analysis should include cell count, protein, glucose, CSF/serum glucose ratio, albumin quotient, IgG index and synthesis rate, oligoclonal bands, broad viral studies including HSV1/2 and VZV PCR and serology, bacterial and fungal cultures if appropriate, cytology, flow cytometry, and an autoimmune encephalopathy/encephalitis panel [25]. Serum MOG and AQP-4 antibodies should be investigated because they can coexist with anti-NMDAR antibodies, and the research of serum MOG antibodies is more sensitive than CSF research [12]. Furthermore, CSF oligoclonal bands can be positive in some forms of autoimmune encephalitis (NMDAR, GABA-B, and GAD encephalitis). To assess the effectiveness of the treatment and rule out a subclinical status epilepticus which can occasionally be present in autoimmune encephalitis patients, an EEG should always be performed [26]. Indeed, a new-onset convulsive or nonconvulsive refractory status epilepticus (NORSE) is typically caused by the presence of an autoimmune encephalitis [26]. Even with a negative brain MRI or minor cortical or subcortical contrast enhancement, the EEG may commonly show a slow and chaotic activity, lateralized periodic discharges, and/or severe delta brush [7,23]. A normal EEG pattern does not exclude autoimmune encephalitis but supports the diagnosis of primary psychiatric disorders [20]. The EEG is abnormal in over 90% of children with NMDARe [12]. However, a normal EEG could be observed in affected patients [23]. We suggest a diagnostic approach for autoimmune encephalitis (Figure 1). In the presence of an autoimmune encephalitis and anti-NMDAR IgG antibodies in CSF, a concomitant cancer must be ruled out. When NMDARe is suspected, a key step in female patients is represented by screening for an ovarian teratoma, using an abdominal or transvaginal US and, if necessary, an abdominal MRI and/or CT [8]. Despite the rare occurrence, male patients should be investigated for testicular cancer by the US and MRI techniques [27].

### 4.1. Brain MRI Features

Almost half of NMDARe patients have aberrant brain MRI findings, detailed in Table 1, which aims to be a quick reference for brain imaging in NMDARe patients [28,29]. In NMDARe, inflammatory infiltrates are commonly observed in the hippocampus. On the other hand, the number of NMDARs in the hippocampus is usually reduced [2]. The brain MRI in patients with NMDARe usually shows a widespread bilateral lesion pattern. Even though inflammation most often affects the hippocampus, abnormalities in the frontal and temporal lobes have also been frequently observed in patients with NMDARe [2,28,30]. The presence of bilateral limbic inflammation is considered the only pathognomonic brain MRI feature that can unquestionably lead to the diagnosis of this autoimmune encephalitis, even though an initial negative brain MRI can result [19,22]. On the other hand, meningeal enhancement, cortical diffusion restriction, and focal or extensive demyelination are uncommon findings in NMDARe [1,7,30]. Many studies have evaluated the relationship between brain MRI lesions and seizures in affected patients [28]. Cortical abnormalities are positively correlated with refractory seizures [31]. Notably, patients with aberrant brain MRI lesions are more likely to have focal seizures than those with normal brain imaging [28]. In patients with autoimmune encephalitis, a normal brain MRI at presentation has been correlated with seizure remission after 6 months of follow-up, although no association is mostly reported between MRI abnormalities and disease prognosis [28,32,33,34,35,36]. Furthermore, postencephalitic epilepsy has been found to be strictly related to brain MRI abnormalities [37]. However, very little is known regarding the relationship between brain MRI and clinical presentation, and the associations between MRI findings and seizure outcomes have not yet been fully elucidated [28].

### 4.2. The Importance of Ultrasonography in NMDARe

The pelvic US represents the primary imaging modality for evaluating the presence of an ovarian mass in a female patient with NMDARe [38]. The presence of a classical mature cystic teratoma is variable, but many different classical ultrasonographic features have been described. Mature teratomas often share common US characteristics, mainly appearing as cystic lesions with echogenic masses due to the presence of hair within the mass and an echogenic Rokitansky nodule, leading to acoustic shadowing (Table 2) [39,40,41]. In addition, mature teratomas are characterized by fluid–fluid levels and by the difference in echogenicity between water and sebum [39]. Many US signs have been associated to mature cystic teratomas, such as echogenic areas with posterior acoustic shadowing (“tip of the iceberg” sign), fat–fluid levels, hyperechoic lines and dots (“dot–dash”), floating spheres (“meat balls sign”), and a hyperechoic, avascular mass (Figure 2) [16,38,39,42,43]. The “tip of the iceberg” sign is related to an echogenic focus in the mass, composed of fat, hair, and cellular debris, which often covers the true extension of the lesion [39,42]. The “dot–dash” sign has a positive predictive value of 98% for mature ovarian teratomas, resulting from hyperechoic dashes and dots due to the presence of hair in the mass with different orientations [16,39]. Instead, the “meat balls/floating balls” sign is related to hyperechoic globules in the cystic mass, composed of keratin, sebum, and hair, which typically change localization with the movement of the patient [39,43]. Meanwhile, immature teratomas manifest as a unilateral heterogeneous solid to cystic masses, often with the presence of sebum and calcifications as solid components [39,44]. On an US examination, calcifications are widespread and appear hyperechoic, not confined within the Rokitansky nodule, while sebum appears as numerous widespread hyperechoic regions. The abdominal MRI and CT are imaging techniques less commonly used for teratoma investigation [39,45]. However, the CT density and MRI scan intensity of teratomas are highly dependent on the presence of the various components, including fat, water, hair, sebum, and calcifications (Figure 3) [39]. The Society of Radiologists in Ultrasound (SRU) published the first consensus statement for the management of asymptomatic ovarian and adnexal cysts detected on the pelvic US [46]. A recent retrospective study showed high sensitivity (100%), specificity (90%), negative predictive value (100%), and accuracy (90%) for the SRU guidelines in detecting ovarian malignancy, while other authors report lower diagnostic rates [39,47,48]. Table 2 could represent a good tool for US features in NMDARe with underlying teratomas.

## 5. Treatment

Several retrospective studies have shown that early and aggressive immunotherapy is associated with a better outcome [11,49]. However, empiric antimicrobial therapy is recommended until infections are excluded. The common practice is to start intravenous antibiotics to cover bacterial meningitis/encephalitis and intravenous acyclovir for HSV/VZV that could be discontinued after the CSF study results are negative [20]. Once infectious aetiologies have been excluded based on the CSF cell count, glucose, and gram stain, immunotherapy should be started to avoid delays. Patients with autoimmune encephalitis are treated with a first-line therapy with high doses of corticosteroids that have good penetration across the blood–brain barrier and have a broad spectrum of anti-inflammatory activity. Usually, a pulse therapy with methylprednisolone is started at the dosage of 30 mg/kg/day per 3–5 days (maximum 1 g/day) in children, followed by a sustained therapy with oral steroids (e.g., prednisolone, 1–2 mg/kg/day) followed by a tapering in 6–12 months [9]. In adults, the use of methylprednisolone 1 g for 3–7 days is recommended as an initial approach [20]. First-line therapy also includes intravenous immunoglobulins (IVIg) or plasma exchange, often in conjunction with corticosteroids [8,50]. A second-line therapy with rituximab and/or cyclophosphamide is uncommonly required [8,50]. Additional complementary immunotherapies, such as bortezomib, and/or tocilizumab, may be considered in resistant cases [50]. According to the literature, corticosteroids are the most popular therapeutic choice in NMDARe patients, although they may potentially cause an initial worsening of the behavioral or psychiatric symptoms, hampering a timely evaluation of the treatment response [20]. An initial approach with combined first-line therapies may be considered in severe forms, including the association of a high dose of corticosteroids with IVIg at 2 g/kg over 2–5 days or an association with plasma exchange, often with good improvement [51]. Longer or repeated IVIg courses may be continued monthly for 3–6 months, depending on severity and availability [52]. If no satisfying clinical response is noted with first-line therapy within 2–4 weeks, the addition of a second-line immunosuppressive drug may improve the outcome. Second-line treatments are recommended, especially in patients with severe disease, with rituximab preferred over cyclophosphamide. Many protocols for Rituximab are accepted (e.g., 375 mg/m^2^ weekly for 4 weeks or two doses of 1 g, 2 weeks apart), and it appears to be a good choice for NMDARe, reducing the risk of relapses [11]. The earlier initiation of second-line therapy seems to be associated with better outcomes compared with late treatment [52]. If the disease does not improve in 1–3 months after the beginning of first-line and second-line therapy, the use of cyclophosphamide can be considered at a dosage of 500–1000 mg/m^2^ (maximum 1500 mg) in monthly pulses for up to 6 months. However, a primary tumour resection is the most important therapeutic step in paraneoplastic NMDARe [10,53,54]. Children with no teratoma-associated encephalitis have a less favorable response to therapy. However, more than 75% of all patients have a good recovery, which is associated with a decline in antibody titres [8]. In a large retrospective study focusing on NMDARe approaches and outcomes, almost all patients were treated with a tumor resection and first-line medical therapy, and half of the patients improved within the first month, with similar response rates in both adults and children. Finally, approximately 80% of patients achieved a good outcome after 48 months, but 12% of them relapsed within the same time [11]. According to a recent review of the literature, the majority of patients with (150/155) who underwent surgical intervention had a favorable disease course [55]. Laparoscopic teratoma removal is widely performed on patients with NMDARe [21]. Encephalitis with no associated tumor and adolescent onset are considered the two most important risk factors for relapse, which is commonly observed in 15–24% of patients, even after several years [56].

## 6. Conclusions

NMDARe is a rare autoimmune disease that frequently affects young women and presents with severe neuropsychiatric involvement. It is important to investigate different causes of encephalitis, despite the fact that the differential diagnosis may be hard. Due to their importance in the diagnostic process, the CSF analysis and brain MRI are currently considered key steps for a correct work-up in NMDARe patients. An ovarian teratoma is commonly found in young female patients and its presence should always be investigated. The abdominal US represents a key imaging technique in the assessment and correct work-up of suspected patients, especially in female adolescents with suggestive manifestations. Imaging investigations play a key role in the diagnosis and outcomes of NMDARe when a concomitant tumor is present, such as an ovarian teratoma. Of these, the US unquestionably represents the primary step due to its affordability, cheapness, and safety.

## Figures and Tables

**Figure 1 brainsci-13-00662-f001:**
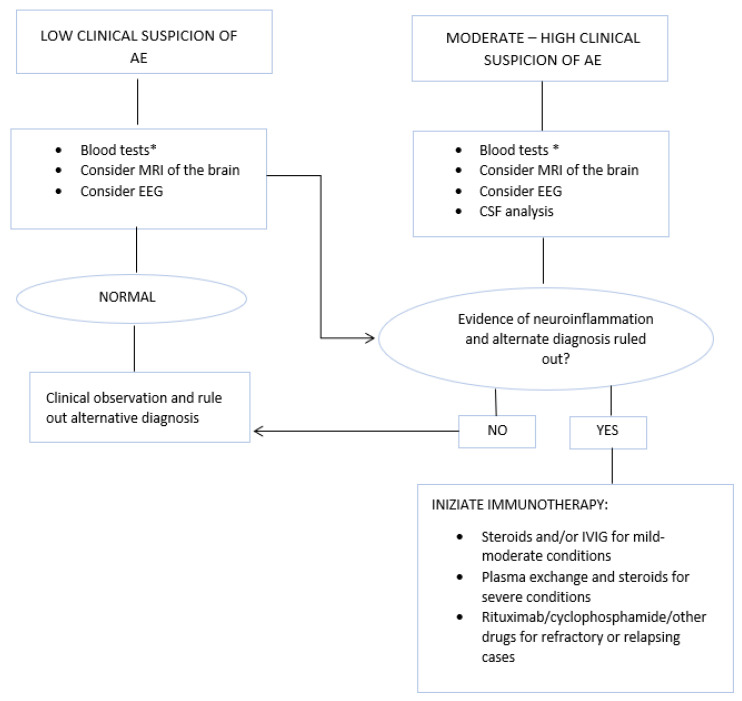
A diagnostic approach for autoimmune encephalitis. * Blood tests should include infectious diseases (erythrocyte sedimentation rate, C-reactive protein, complete blood cell count, and common viruses involved in pediatric encephalitis), neuroinflammatory studies (autoantibodies involved in common pediatric encephalitis, anti-myelin oligodendrocyte glycoprotein antibodies, anti-aquaporin-4 antibodies, and oligoclonal bands), neurorheumatologic studies (angiotensin converting enzyme, anti-nuclear antibody testing, anti-neutrophil cytoplasmic antibody testing, and double-stranded DNA testing), metabolic and mitochondrial testing (lactate/pyruvate ratio, comprehensive metabolic panel, plasma amino acids, ammonia level, copper, ceruloplasmin, vitamin B12, and vitamin B1), and thyroid studies (thyroid stimulating hormone, thyroxine, anti-thyroglobulin antibodies, and anti-thyroid peroxidase antibodies). AE: Autoimmune encephalitis, MRI: Magnetic Resonance Imaging, EEG: electroencephalography, and CSF: cerebrospinal fluid.

**Figure 2 brainsci-13-00662-f002:**
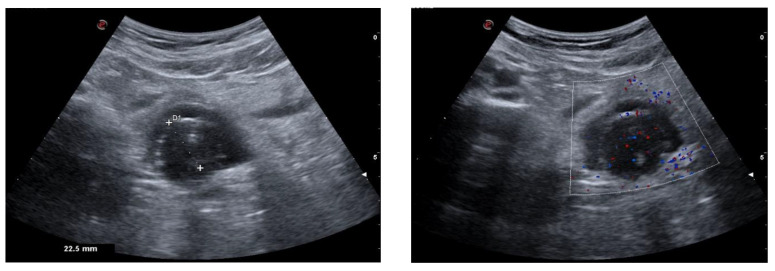
US transverse scan in a female adolescent with NMDARe showing a poor vascularized right ovarian teratoma of about 40 × 22 mm with a well-defined contour and irregular hyperechoic areas inside. D1 is the longitudinal diameter of the teratoma. In the right figure blue and red dots represent the poor vascularization of the ovarian teratoma.

**Figure 3 brainsci-13-00662-f003:**
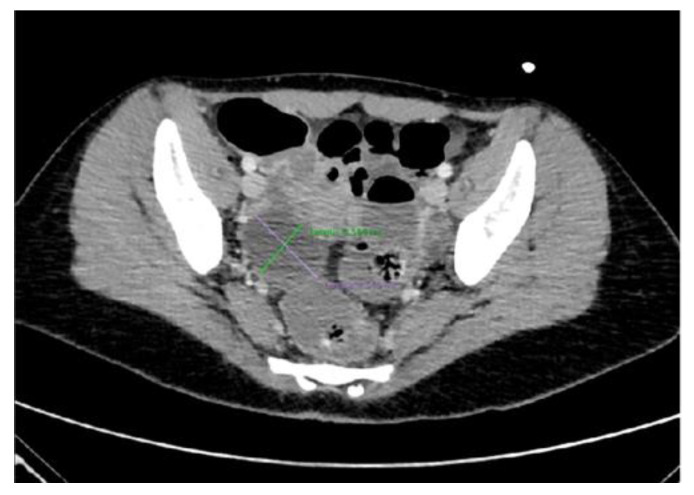
Enhanced CT scan with contrast; portal venous phase: right ovarian teratoma of 43 × 33 mm (indicated by colored lines in the image) with a well-defined contour in a female adolescent with NMDARe. Predominately cystic attenuation with scattered regions of hyperattenuating and fatty tissue attenuation.

**Table 1 brainsci-13-00662-t001:** Brain MRI findings in NMDARe [2,7,28,29,30,31].

Normal Imaging in about Half of Patients
Mostly bilateral brain inflammation
Commonly seen bilateral limbic inflammation, especially in hippocampus but also in cingulate and insula
Less common bilateral involvement of frontal and temporal lobes
Meningeal enhancement, cortical diffusion restriction, and focal or extensive demyelination are rarely present
Cortical inflammation seems positively correlated with refractory seizures, and patients with MRI abnormalities have more focal seizures than patients with normal brain MRI
The relationship between normal brain MRI at onset and seizure remission is controversial

**Table 2 brainsci-13-00662-t002:** Common US findings in mature ovarian teratomas [16,38,39,40,41,42,43,44,45,46,47].

US Finding	Anatomic Counterpart
Single cystic lesion with an echogenic nodule that leads to acoustic shadowing	Unilocular cyst with various septa and a prominent raised protuberance called “Rokitansky nodule”, containing calcifications
“Tip of the iceberg” sign, due to markedly echogenic areas with posterior acoustic shadowing	A marked acoustic shadowing masks the true extent of the teratoma due to presence of an echogenic focus, resulting from the presence of hair, cellular debris, fat, teeth, and calcifications
“Comet tail” sign	Shadowing without an echogenic focus at the tip, due to the presence of hair balls
Fat–fluid and fluid–fluid levels	Different echogenicity between liquid fluid, fat, and sebum
“Dot–dash” sign due to hyperechoic lines and dots	Different orientation of floating hair within the cyst, appearing as dots when perpendicular to the imaging plane, and dashes when parallel
Highly echogenic avascular mass	Poor vascularization and high presence of fat, hair, and calcifications
“Meat/floating balls” sign	Floating hyperechoic balls, composed of sebum, keratin, and hair balls

## Data Availability

No data were generated for this article.

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
