# Peer review of "The Leading Role of Brain and Abdominal Radiological Features in the Work-Up of Anti-NMDAR Encephalitis in Children: An Up-To-Date Review"

_brainsci, 2023, doi:10.3390/brainsci13040662_

Round 1
Reviewer 1 Report
Authors present a review on role of pelvic ultrasound in diagnosis of anti-NDMAR encephalitis in Children.
1. One major issue with this manuscript is that it does not define which type of review this is - narrative, scoping, literature systemic etc?
2. Which literature was used and why? What were the inclusion/exclusion criteria?
3. The title is also somewhat misleading - the method is used for ovarian theratoma, and patient who has teratoma may develope NMDAR encephalitis;. A review on teratomas as well as other forms , i.e. differential diagnosis of NDMAR should be included.
Moderate corrections.
Author Response
Response to Reviewer 1 Comments
- One major issue with this manuscript is that it does not define which type of review this is - narrative, scoping, literature systemic etc?
Response 1: Thank you for your comment. This is a narrative review, and we have stated it into the introduction (line 46).
- Which literature was used and why? What were the inclusion/exclusion criteria?
Response 2: Thank you for your comment. In order to focus on the radiologic characteristics of anti-NMDAR encephalitis, we mostly opted for reviews that emphasized the use of pelvic ultrasound and brain MRI in the correct diagnostic work-up. We sought to utilize the most recent papers available on PubMed, although also some older papers have also been included due to their clarity and thoroughness. The authors' intention was to provide a clear narrative review; thus inclusion and exclusion criteria have not been specified into the text.
- The title is also somewhat misleading - the method is used for ovarian theratoma, and patient who has teratoma may develope NMDAR encephalitis;. A review on teratomas as well as other forms , i.e. differential diagnosis of NDMAR should be included.
Response 3: Thank you for your suggestion. We have modified the title of the review, highlighting that the aim of this paper is to provide a detailed description of the radiological features of anti-NMDAR encephalitis rather than to offer a complete differential diagnostic tool to other forms of encephalitis. However, a diagnostic and therapeutic algorithm has been added to enrich the quality of the paper.
Reviewer 2 Report
A very interesting review. Despite the aim of the authors was to describe the usefulness of pelvic ultrasound in the etiological diagnosis of NMDAR encephalitis they reviewed the clinical diagnostics, and therapeutics issues of this entity. The review could enhance by adding some tables or figures with the diagnosis and treatment algorithms that they described in the text.
Author Response
Response to Reviewer 2 Comments
- A very interesting review. Despite the aim of the authors was to describe the usefulness of pelvic ultrasound in the etiological diagnosis of NMDAR encephalitis they reviewed the clinical diagnostics, and therapeutics issues of this entity. The review could enhance by adding some tables or figures with the diagnosis and treatment algorithms that they described in the text.
Response 1: Thank you for your comment. We have added a figure that aims to provide a clear algorithm for both diagnosis and treatment approaches.
Reviewer 3 Report
The manuscript requires major changes to be considered in BrainSciences.
a) The manuscript contains elements as a review but also as a case-report. Please adjust to improve presentation.
b) There are many recent reports of NMDA-encephalitis in relationship with teratoma (with p-USG). Please revise the content; this could enrich the section 4 of your manuscript.
Liu, Y., Tian, Y., Guo, R., Xu, X., Zhang, M., Li, Z., ... & Huang, X. (2022). Anti-NMDA Receptor Encephalitis: Retrospective Analysis of 15 Cases, Literature Review, and Implications for Gynecologists. Journal of Healthcare Engineering, 2022.
Li, S. J., Yu, M. H., Cheng, J., Bai, W. X., & Di, W. (2022). Ovarian teratoma related anti-N-methyl-D-aspartate receptor encephalitis: A case series and review of the literature. World Journal of Clinical Cases, 10(16), 5196.
Begum, J., Aziz, Z., Sahoo, S. K., Majumder, R., & Sable, M. N. (2022). Anti-N-Methyl-D-Aspartate receptor encephalitis associated with mature ovarian teratoma in a young adolescent: a case report. Journal of Pediatric and Adolescent Gynecology, 35(3), 400-403.
Kojima, M., Kurihara, S., Saeki, I., Izumo, H., Tateishi, Y., Kobayashi, Y., ... & Hiyama, E. (2022). Paediatric anti-NMDA-receptor encephalitis with ovarian teratoma. Journal of Pediatric Surgery Case Reports, 83, 102318.
Lee, S. H., Lee, C. Y., Park, H. S., Park, J., & Yun, J. Y. (2023). Anti-N-Methyl-D-Aspartate Receptor (NMDAR) Encephalitis Associated With Mediastinal and Ovarian Teratomas: A Case Report. Journal of Korean Medical Science, 38(6).
Nguyen, L., & Wang, C. (2022). Anti-NMDA Receptor Autoimmune Encephalitis: Diagnosis and Management Strategies. International Journal of General Medicine, 7-21.
French, A. V., & Grossmann, L. (2022). Early diagnosis and treatment improve outcomes for premenarchal patients with anti-NMDA receptor encephalitis and ovarian teratoma. Journal of Pediatric and Adolescent Gynecology, 35(4), 516-517.
Malhotra, D., Sane, S., Mane, S., & Hajirnis, O. (2023). Anti-N-methyl-D-aspartate receptor encephalitis: A paraneoplastic syndrome in an Indian adolescent girl. Indian Pediatrics Case Reports, 3(1), 39.
The content of tables should be clearly supported for specific articles, but also it should be presented as a tool for US-MRI diagnosis in these cases.
Title could be improved to attract readers due to the multiple reviews and recent case reports (with literature review).
The introduction should highlight the relevance of Brain disruption. The article must be clearly into the BrainSciences aims.
Minor editing seems to be required
Author Response
Response to Reviewer 3 Comments
- a) The manuscript contains elements as a review but also as a case-report. Please adjust to improve presentation.
Response 1: Thank you for your suggestion. We have slightly modified the legend of the Figure 2. However, we wanted to enrich this paper with novel images from a patient with NMDARe and an underlying right ovarian teratoma. We sought to provide both detailed tables and findings from the literature and also clear radiographic images from a real pediatric patient. The authors are fully available to remove these images and merely focus on the evidence from literature, if this is the reviewer’s opinion.
- b) There are many recent reports of NMDA-encephalitis in relationship with teratoma (with p-USG). Please revise the content; this could enrich the section 4 of your manuscript.
Liu, Y., Tian, Y., Guo, R., Xu, X., Zhang, M., Li, Z., ... & Huang, X. (2022). Anti-NMDA Receptor Encephalitis: Retrospective Analysis of 15 Cases, Literature Review, and Implications for Gynecologists. Journal of Healthcare Engineering, 2022.
Li, S. J., Yu, M. H., Cheng, J., Bai, W. X., & Di, W. (2022). Ovarian teratoma related anti-N-methyl-D-aspartate receptor encephalitis: A case series and review of the literature. World Journal of Clinical Cases, 10(16), 5196.
Begum, J., Aziz, Z., Sahoo, S. K., Majumder, R., & Sable, M. N. (2022). Anti-N-Methyl-D-Aspartate receptor encephalitis associated with mature ovarian teratoma in a young adolescent: a case report. Journal of Pediatric and Adolescent Gynecology, 35(3), 400-403.
Kojima, M., Kurihara, S., Saeki, I., Izumo, H., Tateishi, Y., Kobayashi, Y., ... & Hiyama, E. (2022). Paediatric anti-NMDA-receptor encephalitis with ovarian teratoma. Journal of Pediatric Surgery Case Reports, 83, 102318.
Lee, S. H., Lee, C. Y., Park, H. S., Park, J., & Yun, J. Y. (2023). Anti-N-Methyl-D-Aspartate Receptor (NMDAR) Encephalitis Associated With Mediastinal and Ovarian Teratomas: A Case Report. Journal of Korean Medical Science, 38(6).
Nguyen, L., & Wang, C. (2022). Anti-NMDA Receptor Autoimmune Encephalitis: Diagnosis and Management Strategies. International Journal of General Medicine, 7-21.
French, A. V., & Grossmann, L. (2022). Early diagnosis and treatment improve outcomes for premenarchal patients with anti-NMDA receptor encephalitis and ovarian teratoma. Journal of Pediatric and Adolescent Gynecology, 35(4), 516-517.
Malhotra, D., Sane, S., Mane, S., & Hajirnis, O. (2023). Anti-N-methyl-D-aspartate receptor encephalitis: A paraneoplastic syndrome in an Indian adolescent girl. Indian Pediatrics Case Reports, 3(1), 39.
Response 2: Thank you for your suggestion. We have added relevant information from these interesting articles (evidenced phrases).
- The content of tables should be clearly supported for specific articles, but also it should be presented as a tool for US-MRI diagnosis in these cases.
Response 3: Thank you for your comment. We have presented tables 1 and 2 as quick and relevant tools for the correct interpretation of both abdominal US and brain MRI findings in NMDARe patients. In addition, the correct references have been specified in the figure legends.
- Title could be improved to attract readers due to the multiple reviews and recent case reports (with literature review).
Response 4: Thank you for your comment. We have modified the title of the paper, as you suggested.
- The introduction should highlight the relevance of Brain disruption. The article must be clearly into the BrainSciences aims.
Response 5: Thank you for your suggestion. We have added a relevant statement regarding the common brain disruption in patients affected by NMDARe in the introduction of the paper (line 44).
Round 2
Reviewer 1 Report
Authors have sufficiently responded to remarks, I suggest this narrative review for publication.
Acceptable.
Author Response
Thank you very much for your comment
Reviewer 3 Report
The references in Table 1 and Table 2 should be clearly linked to specific data.
It is OK! minor details
Author Response
Response to Reviewer 3 Comment
- The references in Table 1 and Table 2 should be clearly linked to specific data.
Response 1: Thank you for your suggestion. The references in Tables 1 and 2 have been updated to provide a more clear link to specific data.